# Enhancing the Recognition Task Performance of MEMS Resonator-Based Reservoir Computing System via Nonlinearity Tuning

**DOI:** 10.3390/mi13020317

**Published:** 2022-02-18

**Authors:** Jie Sun, Wuhao Yang, Tianyi Zheng, Xingyin Xiong, Xiaowei Guo, Xudong Zou

**Affiliations:** 1The State Key Laboratory of Transducer Technology, Aerospace Information Research Institute, Chinese Academy of Sciences, Beijing 100190, China; sunjie17@mails.ucas.ac.cn (J.S.); zhengtianyi17@mails.ucas.ac.cn (T.Z.); guoxaiowei16@mails.ucas.ac.cn (X.G.); 2School of Electronic, Electrical and Communication Engineering, University of Chinese Academy of Sciences, Beijing 100049, China; yangwh@aircas.ac.cn (W.Y.); xiongxy@aircas.ac.cn (X.X.); 3QILU Aerospace Information Research Institute, Jinan 250101, China

**Keywords:** nonlinear dynamics, reservoir computing, micromechanical resonator, pattern recognition

## Abstract

Reservoir computing (RC) is a potential neuromorphic paradigm for physically realizing artificial intelligence systems in the Internet of Things society, owing to its well-known low training cost and compatibility with nonlinear devices. Micro-electro-mechanical system (MEMS) resonators exhibiting rich nonlinear dynamics and fading behaviors are promising candidates for high-performance hardware RC. Previously, we presented a non-delay-based RC using one single micromechanical resonator with hybrid nonlinear dynamics. Here, we innovatively introduce a nonlinear tuning strategy to analyze the computing properties (the processing speed and recognition accuracy) of the presented RC. Meanwhile, we numerically and experimentally analyze the influence of the hybrid nonlinear dynamics using the image classification task. Specifically, we study the transient nonlinear saturation phenomenon by fitting quality factors under different vacuums, as well as searching the optimal operating point (the edge of chaos) by the static bifurcation analysis and dynamic vibration numerical models of the Duffing nonlinearity. Our results in the optimal operation conditions experimentally achieved a high classification accuracy of (93 ± 1)% and several times faster than previous work on the handwritten digits recognition benchmark, profit from the perfect high signal-to-noise ratios (quality factor) and the nonlinearity of the dynamical variables.

## 1. Introduction

Currently, artificial neural networks (ANN) [1] are widely used in the emerging field of artificial intelligence (AI) to process a considerable amount of information that is generated by various terminal sensors all the time. Inspired by the way the human brain works, recurrent neural networks (RNNs) [2,3,4] have emerged as one of the most powerful neuromorphic computing paradigms to solve complex time-dependent tasks. However, the complex and time-consuming training algorithms to train the connection weights between the nodes make it computationally expensive and difficult to implement on hardware. The reservoir computing (RC) [5] paradigm provides a solution to this by using an RNN with fixed connection weights (the reservoir) to transform inputs into a higher dimensional representation prior to them being passed to a linear readout layer with trainable weights. The core features inherited from recurrent neural networks make them suitable for temporal information processing. Moreover, time multiplexing allows it to be performed even if the reservoir consists of only a single dynamical node. Virtually any device or material exhibiting sufficient nonlinear dynamics and fading memory characteristics can be used as a physical reservoir. Such implementations have the potential for lower energy costs than traditional software implementations of RC (echo state networks [6,7] or liquid state machines [8]) as they can directly utilize the intrinsic characteristics of the physical systems. This has led to a wide range of systems being proposed as suitable reservoirs, such as electronic devices [9,10], optoelectronics [11,12,13,14,15], spintronic devices [16], memristors [17,18,19,20,21,22], and micromechanical resonators [23,24,25].

Particularly, micromechanical resonator-based RC combines the advantages of the micro-electro-mechanical system (MEMS) devices [26,27] and the physical RC [28,29], such as small size, low consumption, compatibility with CMOS technology or MEMS sensors (MEMS accelerometers, MEMS pressure sensors, and so on), rendering it convenient to process sensing signals in the analog domain directly (especially signal identification and classification), and greatly reduces the amount of redundant terminal data and improves the security of information. Very recently, a single silicon beam resonator device was first proposed to achieve RC using a classical Duffing nonlinearity as the source of nonlinearity and was found to have huge potential applications in combining the functions of sensing and computing [23]. However, these studies mainly focused on the pioneering realization of new RC hardware such as single beam resonators and neural accelerometers [23,24], while the in-depth exploration of the relationship between the nonlinear mechanism of MEMS devices and the RC performance is lacking. In our previous study, we first proposed a novel RC architecture using a single hybrid nonlinear (HNL) resonator [25] and verified its competitive performance through different tasks. Furthermore, we originally demonstrated that the transient nonlinearity affects the memory capacity of the reservoir and part of the nonlinear mapping ability at the same time, while the duffing nonlinearity of the resonator provides the main nonlinear mapping function. The combination of the two nonlinearities (HNL) achieved high recognition accuracy and increased the processing speed by hundreds of times compared with the original time-delay feedback RC architecture.

Here, we propose a hybrid nonlinear combinatorial modulation strategy to search for the optimal operating condition to achieve higher accuracy and faster computing speed for the hardware RC we have presented in our previous work. Furthermore, we study the physical mechanism of the influence of two nonlinearities in-depth by constructing experimental and simulation models. Meanwhile, we study the transient nonlinear saturation phenomenon by fitting quality factors under different vacuums, as well as the static bifurcation analysis in the Hamiltonian system and the dynamics vibration analysis according to the numerical model in the Duffing nonlinearity. The results demonstrate that the best operating point is located at the edge of chaos [23,30]. Significantly, this work may offer an optimization method to improve the computing properties of hardware RC using MEMS devices with similar principles.

## 2. Hardware RC Implementation

The clamped–clamped silicon beam micro-resonator was fabricated on a (100) p-doped silicon on glass substrate by the standard Silicon on Glass (SOG) process. The device layer thickness of *T* = 40 μm defines the width of the beam, the electrode length is *Le* = 360 μm, and the length, in-plane thickness, and the gap between the beam and the drive/sense electrode were chosen to be *L* = 500 μm, *W* = 6.5 μm, and *d* = 3 μm, respectively. Essentially, electrostatically actuated MEMS resonators may exhibit rich dynamics, and the parallel plate drive and detection modes are more prone to complex nonlinearity than comb drive and detection modes. The actual device SEM diagram of the designed resonator is shown in Figure 1a, and its displacement can be approximated by the Duffing nonlinear equation [26]:(1)$$m\ddot{z}+c\dot{z}+k_{1}z+k_{3}z^{3}=\frac{1}{2}\frac{C_{0}d}{{\left(d-z\right)}^{2}}{\left(V_{dc}+V_{ac}\sin\Omega t\right)}^{2}-\frac{1}{2}\frac{C_{0}d}{{\left(d+z\right)}^{2}}{\left(V_{dc}\right)}^{2}\;$$
where z, $\dot{z}$, and $\ddot{z}$ are the displacement, velocity, and acceleration of the resonator, respectively, $w_{n}=\sqrt{k_{1}/m}$ is the natural angular frequency of the resonator in its linear regime, *m* is the effective mass of the beam, c is damping term, $k_{1}$ is linear stiffness term, $k_{3}$ is Duffing nonlinear stiffness term, $k_{1}=\frac{32ETW^{3}}{L^{3}}$*,* $k_{3}=\frac{k_{1}}{\sqrt{2}W^{3}}$, *E* is the Young’s modulus of silicon, $C_{0}$ and *d* are the initial capacitance and gap of the parallel plate, $C_{0}=\frac{\epsilon_{0}\times Le\times T}{d}\approx 4.307e-14\;F,$ $\epsilon_{0}$ is the permittivity of vacuum, $V_{dc}$ is the bias voltages, and $V_{ac}$ and Ω are the driving amplitude and frequency.

As shown in Figure 1a, the designed micro-resonator was put on a vacuum adjustable sealed chamber to change the air pressure for different quality factors (*Q*). Before being supplied to the drive electrode, the input signal is preprocessed in the digital domain; the LABVIEW (LABVIEW 2019, National Instruments, Texas, United States) program controls 12-bits NI 6366 X Series Data Acquisition (DAQ) (National Instruments, Texas, United States) to realize a conversion of digital to analog, which modulates a sinusoidal drive of amplitude and frequency. Simultaneously, the resonator response is measured by MEMS interface circuit, then digitized by DAQ, detecting its envelope and down-sampling, before saving it and triggering the next sample loop. Figure 1b shows an example of a resonator motion driven by the random amplitude modulated sine wave signal with an interval time of *θ* = 6.25 ms; it depicts the hybrid nonlinear interaction mechanism between neurons, which was described in [25]. The best performance of the HNL-RC can be tuned by the nonlinear tuning (transient nonlinear and Duffing nonlinear), the transient nonlinear function $x\left(t\right)=x_{0}\pm Β\times e^{-\frac{w_{n}}{2Q}t}$ can be derived by the typical linear underdamped second-order oscillation system, its nonlinear strength mainly limited with quality factor *Q* when fixed the natural angular frequency $w_{n}$. The Duffing nonlinear characteristic response is very sensitive to the choice of the reference operating points, which are normally determined by testing the amplitude hysteresis curve and frequency hysteresis curve shown in Figure 1c,d. Finally, the study of nonlinear physical properties and tuning theory are beneficial to build a better HNL resonator reservoir that will be proved by the handwritten digit recognition task below.

We performed a benchmark task called handwritten digit recognition that is common in reservoir computing for hardware implementations. The Mixed National Institute of Standards and Technology (MNIST) database [31] was created by “remixing” the digit samples written by high school students and employees of the United States Census Bureau. Each sample in the dataset was composed of a 28 × 28 gray value matrix. We randomly selected 1000 samples from the original training database, which contained 100 samples for each of the digits “0–9” (900 samples as a training set, 100 samples as a test set). The chosen part of the MNIST handwritten digit database contained 1000 grey images that we called the GMNIST dataset, and BMNIST is binary images dataset transformed from GMNIST for the transient nonlinear analysis.

A basic RC system processing of the pattern recognition was performed in the reservoir, as shown in Figure 2a. The original grayscale image 28 pixels × 28 pixels was trimmed to a 22 pixels × 20 pixels image for reducing redundant information. Then, the 22 pixels × 20 pixels matrix was transformed into 1 × 440 temporal sequences of input pulse streams with separation time *θ*, and the driver carrier signal was multiplied at a certain frequency and then placed into the HNL reservoir, which is constructed by 440 “neural” nodes. Finally, we obtained a 440 × 10 readout network that was used for classification after training. In order to perform the MNIST classification function, ten appropriate target functions were constructed as ten linear classifiers; each is a polynomial function composed of the optimal weight coefficient vector w, and to prevent the overfitting caused by excessive feature information during the training stage, we introduced ridge regression with Tikhonov regularization (L2 norm) to solve this problem and obtain the minimum mean square error between predicted value y and target value $y_{t}$ by adjusting the regularization coefficient λ.
(2)$$\{\begin{array}{c} y_{i}\left(t\right)=w_{i}^{T}x\left(t\right)\\ \;w_{i}=y_{ti}X^{T}{\left(XX^{T}+\lambda I\right)}^{-1} \end{array}$$
where *i* is represented different classifiers coefficient, x(t) is an RC response vector of each sample, X is the data matrix that contains response signals of all samples, and I is a unit matrix.

After the training process, we applied every test sample to the optimal ten classifiers, as shown in Figure 2b. It would select the actual digit through a “winner-takes-all” approach, the predicted results would be picked up from the 10 classifier output results by the maximum method, then every target function is +0 if the digit does not correspond to the sought digit, and +1 if it does. Finally, we obtained the recognition accuracy after comparing it with the target value, and the final accuracy is the average after ten-order cross-validation.

## 3. Transient Nonlinearity Tuning

The principle of HNL-RC networks regulation changes due to the introduction of transient nonlinearity, which exists in the oscillation starting and decay stages of the resonator compared to the time-delay RC only with Duffing nonlinear source. In the HNL-RC, the fading memory and the nonlinear richness would be simultaneously changed through quality factor *Q*, due to the decay time function $T=\frac{2Q}{w_{n}}$ and the transient nonlinear function; it is different from the time-delay RC that must be adjusting the length of the mask (*M*), feedback factor value, and feedback loops under fixed a decay time *T*. Hence, all the above analysis points to the kernel parameter, *Q*, which mainly determines the RC system’s computation speed and part of the nonlinear mapping capability, so this property is essential to study and optimize for recognizing and processing temporal sequences.

### 3.1. Q Value Fitting Model

Conventional quality factor measurement methods include the −3 dB bandwidth method and ring-down method, etc. [32,33]. However, the measurement condition is generally linear vibration state, which is not completely suitable for the nonlinear vibration state. Here, we used a sweep frequency data (SFD) fitting model to estimate *Q* that is suitable for linear and nonlinear vibration states. For theoretical analysis, the electrostatic drive term (Formula (1)’s right side) can be simplified by Taylor expansion, and the third term can be retained and rewritten as Formula (3), then the detailed derivation process of the SFD Formula (4) can be seen [32].
(3)$$m\ddot{z}+c\dot{z}+K_{1}z+K_{3}z^{3}=F_{act}\;\sin\;\Omega t$$
where $K_{1}=k_{1}-k_{e1},$ $K_{3}=k_{3}-k_{e3},\;\;k_{e1},\;k_{e3},\;F_{act}$ are the linear, cubic nonlinear electrical terms, and magnitude of the forcing term at frequency Ω, respectively,$\;k_{e1}=\frac{2C_{0}V_{dc}^{2}}{d^{2}}$, $\;k_{e3}=\frac{3C_{0}}{d}\;k_{e1},\;\;F_{act}=\frac{C_{0}V_{ac}V_{dc}}{md}$.

The results
$R^{2}=0.9715$ and $Q_{estimate}\approx 1383$ using the above transient nonlinear function to fit the by the ring-down method, and $R^{2}=0.9999$, $Q_{estimate}\approx 1051$ using the SFD fitting function in the same simulation setup (*Q* = 1000), which proves the superiority of the SFD fitting method. Figure 1e shows the SFD fitting results for the sweep frequency experiment in a fixed are pressure, and it provides accurate calibration for the following experiments to evaluate the influence of *Q* value on transient nonlinearity.
(4)$$f=f_{n}\left(1+{bx}^{2}-\frac{1}{2Q}\sqrt{\frac{1}{x^{2}}-1}\right)\;$$
where $f_{n}$ is represented the natural frequency of the resonator, $x=\frac{A}{A_{max}}$ is amplitude ratio, and *A* is resonator response amplitude, and *b* is a simplification factor.

### 3.2. Experimental Analysis

In order to avoid the interference of the Duffing nonlinearity on the classification performance of the system as much as possible, we introduced the BMNIST database that is composed of binary data. Firstly, the *Q* value was estimated by the SFD fitting model in different air pressures, and we chose four different *Q* as the verification experiments according to the real test conditions. Figure 3a illustrates the classification accuracy improves from 77% to 89% in the experimental results, which reveals the increased noise and weak nonlinear strength in a low-*Q* environment decrease the system recognition performance, the simulation accuracy also increased by 3% under the same parameters setup as possible without system noise, and more *Q* values are chosen for deeper study. Similar to the saturation conclusion, the accuracy is not improved with a larger *Q* than the optimal value. The histogram distribution about the response digits “7” and “9” is more intuitive. Figure 3b implies that it is easier to distinguish between the optimal *Q* value than lower. The computing speed is several times improved while ensuring the nonlinear richness of the reservoir is unchanged by tuning the *Q* value, which can commonly be used in almost all hardware-RC implementation systems with a similar principle.

## 4. Duffing Nonlinearity Tuning

### 4.1. Experimental Analysis

In the HNL-RC architecture, Duffing nonlinearity is the most essential source of nonlinear richness for pattern classification tasks. Thus, to obtain insight into the optimal operating conditions, we investigated the image recognition systematically in a wide range of the amplitude $V_{ac}$. We turned to evaluate the gain in overall performance provided by the experiments at the optimal *Q*. The GMNIST dataset is used to test Duffing nonlinearity performance due to taking into account the richer nonlinear response triggered by multi-valued information. The classification performance is expressed in two ways: the error rate that shows the percentage of digits that have been wrongly classified, and the t-Distributed Stochastic Neighbor Embedding (t-SNE) technique to represent our reservoir response data in a 2D plot (see Figure 4b) to visualize how the data separation occurs and understand the recognition capacity of the different operation point.

Overall, as shown in Figure 4a, for a wide range of values of $V_{ac}$, the Duffing nonlinearity gradually reduces the classification error rate, except for the point $V_{ac}=1\;V$ due to the input signal data being distributed on the upper and lower branches of the amplitude hysteresis curve so that the correlation between the contexts is de-correlated. An optimum nonlinearity is reached for $V_{ac}=5\;V$ providing the highest classification rate of (93±1)%, which is visualized by t-SNE in Figure 4b. During the data reduction, the probability of two vectors being neighbors is conserved, allowing visualization of the structure in the data. Each digit is represented by colored dots for all data points of the utterances.

### 4.2. Simulation Analysis

In particular, the error rate drastically increased when the $V_{ac}>5\;V$, even the resonator fails. In order to have a deep insight into the resonator vibration, static bifurcation analysis of the Hamiltonian system is carried out to obtain the bifurcation sets. In this theoretical analysis, the dissipation and excitation voltage can be regarded as small perturbations, and the premises are $V_{dc}\gg V_{ac}$ and high *Q* value. Furthermore, some particular phenomena may be overlooked by Taylor’s expansion of the electrostatic force when static bifurcation analysis is carried out. Therefore, the corresponding non-dimensional equation of the Hamiltonian system of Formula (5) [34,35,36] without any approximation would be more appropriate to investigate the whole equilibrium position property and complicated vibration properties of the resonator.
(5)$$\{\begin{array}{c} \dot{x}=y\\ \dot{y}=-\alpha x-\beta x^{3}+\frac{4\gamma x}{1-2x^{2}+x^{4}}+\varepsilon \left(-\mu y+\frac{2\gamma V_{ac}\sin w\tau }{V_{dc}\left(1-2x+x^{2}\right)}\right) \end{array}$$

Introduce the following non-dimensional variables:(6)$$\tau =w_{n}t;\;w=\frac{\Omega }{w_{n}};\;x=\frac{z}{d};\;\alpha =\frac{k_{1}}{m\times w_{n}^{2}}\;;\;\beta =\frac{k_{3}d^{2}}{m\times w_{n}^{2}};\;\gamma =\frac{C_{0}V_{dc}^{2}}{2m\times w_{n}^{2}\times d^{2}}$$

Figure 5a shows the number of abscissa coefficients varies (equilibrium positions) as the variation of $V_{dc}$ under β=0.15 (determined by the resonator size). Here, the static pull-in voltage can be derived by $\gamma =\frac{{\left(1+\beta \right)}^{3}}{27\beta^{2}}$, which detailed analysis in [35] under different β and γ conditions. In the analysis model, the $V_{pull-in}\approx 93\;V$ can be represented as the boundary when the perturbations are assumed to be small enough, the method of multiple scales is generally used to investigate the dynamic nonlinear behavior of the resonator with small $V_{ac}$ for explaining the above experimental results. However, the excitation amplitude to be studied exceeds the boundary condition of theoretical analysis; hence, we constructed the numerical model using Formula (1) to study the changes in nonlinear vibration by the phase portraits. As shown in Figure 5b–e, which illustrates the displacement–acceleration and the displacement–velocity phase portraits for an AC voltage of 5 V and a pull-in AC voltage 5.3 V in the fixed DC bias 30 V, the dynamics system results verify the optimal operating point occurs at the edge of chaos, as observed before.

Detailed numerical studies, including phase portrait, bifurcation diagram, and Poincare map, demonstrate the analytical prediction and reveal the effect of excitation amplitude on the system transition to chaos, and this procedure can be observed more obviously between stable and pull-in states in the special resonator size [37].

## 5. Conclusions

To summarize, we numerically and experimentally studied the mechanism and optimization method of the micromechanical resonator-based RC with hybrid nonlinear dynamics and innovatively introduced a nonlinear tuning strategy to analyze the computing properties (the processing speed and recognition accuracy) of the presented RC. Specifically, we studied the influencing mechanism of transient nonlinearity by fitting quality factors through the SFD equation under different vacuums. Furthermore, we numerically verified its saturated nonlinear characteristics and corresponding optimal values, which increased the processing speed of the system several times. Additionally, both the static bifurcation analysis and dynamic vibration numerical model were used to prove the optimal Duffing nonlinear operating point is the edge of chaos, which is intuitively observed by the phase portraits and t-SNE plots, as well as the high handwritten digits recognition accuracy of (93±1)% experimentally under the optimal conditions. Our results demonstrate that the HNL-RC architecture based on micro–resonator can achieve higher speed information processing capabilities and recognition accuracy by the above strategies of nonlinear tuning, which is very suitable for pattern recognition tasks and pave a way to build a better physical reservoir.

## Figures and Tables

**Figure 1 micromachines-13-00317-f001:**
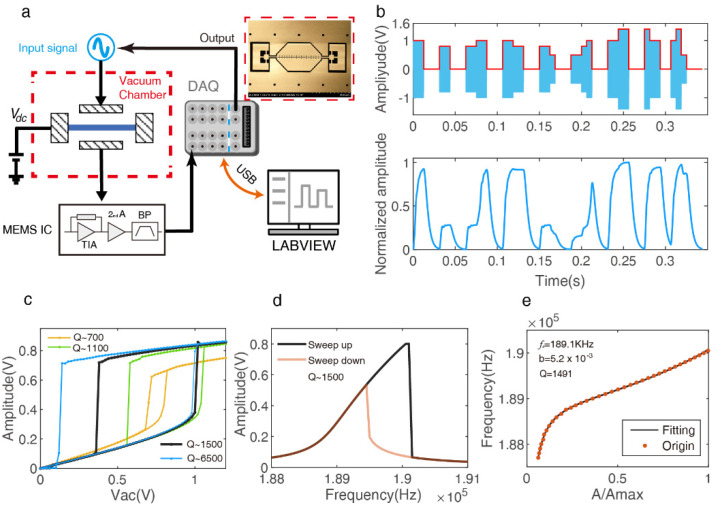
Nonlinear dynamics of the double clamped beam resonator under different stimulation. (**a**) Schematic of the experimental setup of the single hybrid nonlinear (HNL) reservoir computer. (**b**) HNL reservoir states response to different amplitude input signals; it depicts the complex connection between neuron nodes. (**c**) Amplitude sweeping open-loop experiment in different air pressure and a fixed bias voltage of $V_{dc}=30\;V$, to obtain the hysteresis curve about the excitation amplitude and suitable effective $V_{ac}$ range. (**d**) Frequency sweep open-loop experiment for a fixed bias voltage of $V_{dc}=30\;V$, $V_{ac}=1\;V$ and Q≈1500, to obtain the driving frequency $f_{d}=189.5\;\mathrm{KHz}$ at the front bifurcation point. (**e**) Quality factor estimates of the resonator with nonlinear vibrations. It depicts the fitting result of the data on the left side of the resonate frequency point of the “Sweep up” curve in (**d**).

**Figure 2 micromachines-13-00317-f002:**
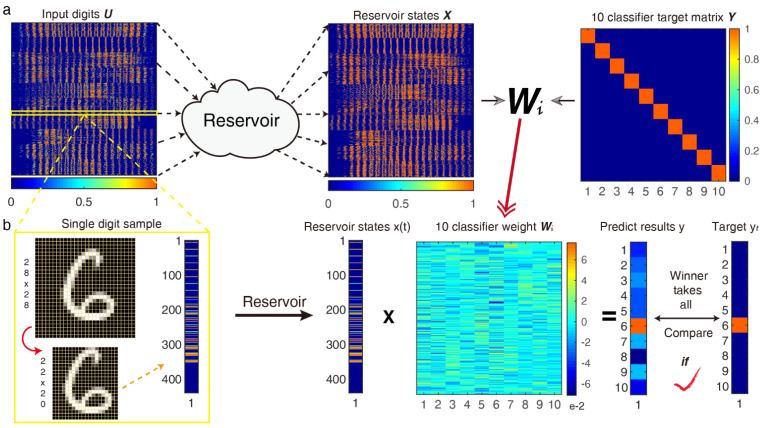
Handwritten digit recognition using an HNL resonator-based RC system. (**a**) Schematic diagram of the training process; put the n×440 (n is the number of samples) input digits matrix to the reservoir in sequence, then obtain the states matrix ***X***, and the ridge regression algorithm is activated to work for the final optimal 10-classifiers weight matrix $W_{i}$; (**b**) Single handwritten digit test process in the test dataset, signal preprocessing is shown in the yellow box.

**Figure 3 micromachines-13-00317-f003:**
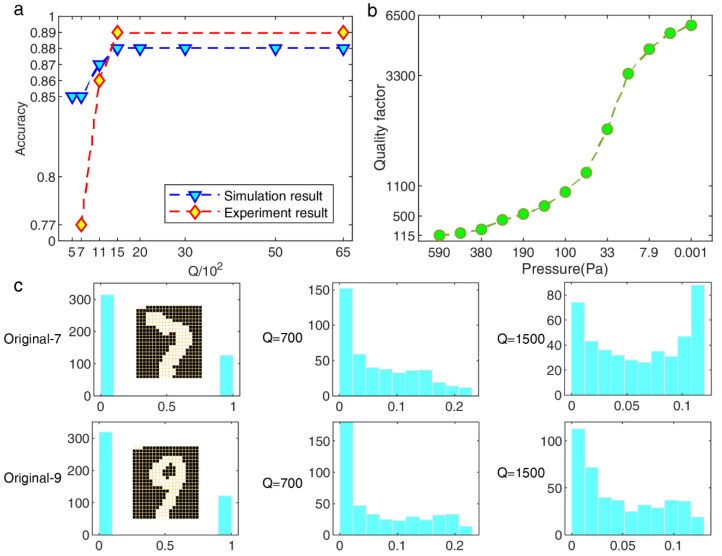
BMNIST classification results in simulation and experimental tests. (**a**) Classification accuracy with different *Q*. The test conditions of the experiment are $V_{dc}=30\;V$ and $V_{ac}=1.5\;V$, $f_{d}=189.5\;\mathrm{KHz}$, the optimal $\theta =\frac{1}{2}T$. (**b**) Calibration results about *Q* vs. Pressure by sweep frequency data (SFD) model; different pressures were adjusted by the vacuum chamber needle valve, and the BMNIST experiment was carried out under four stable pressure conditions chosen from this result. (**c**) Histogram distribution of HNL-RC states output for a single sample.

**Figure 4 micromachines-13-00317-f004:**
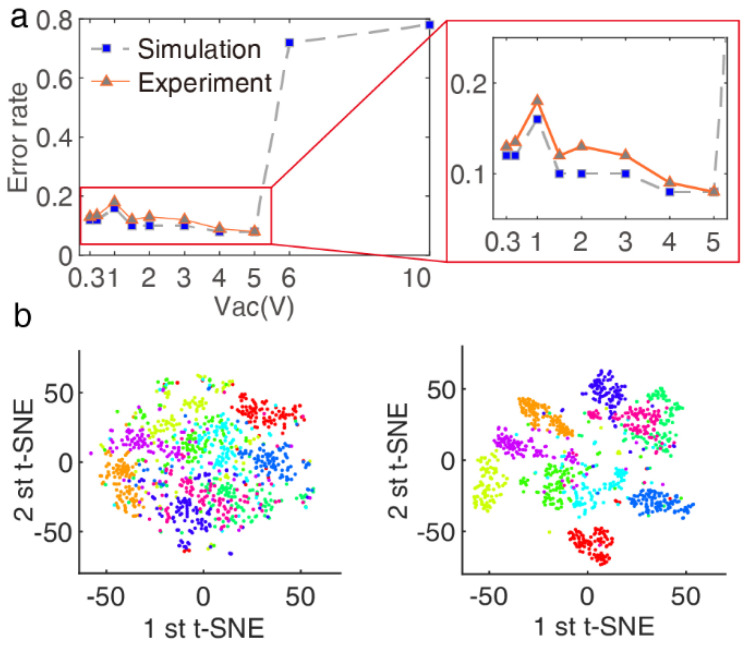
Optimal Duffing nonlinear working interval for handwritten digit recognition. (**a**) Classification error rate results after simulation and experiment by changing the driving voltage $V_{ac}$. The best–recognized accuracy is (93±1)%  when $V_{ac}=5\;V$, $V_{dc}=30\;V$ and $f_{d}=189.5\;\mathrm{KHz}$, Q≈1500. (**b**) Two–dimensional representation of the two t-SNE components: left one with $V_{ac}=1\;V$, right one with $V_{ac}=5\;V$ in (**a**).

**Figure 5 micromachines-13-00317-f005:**
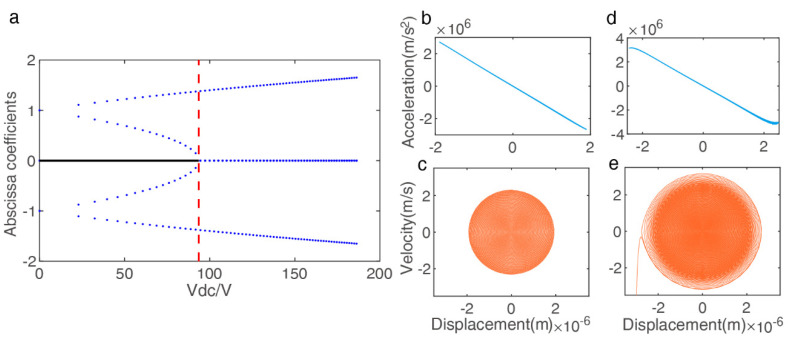
Simulation models to analyze the complex nonlinear oscillations (nonlinear/chaotic region) in the electrostatic resonator. (**a**) Static bifurcation of the Hamiltonian system versus $V_{dc}$ under fixed β (solid line is the one stable center point; points are unstable saddle points), (**b**,**c**) Phase plots of the steady period response at an AC voltage (5 V): no AC symmetry breaking, (**d**,**e**) Phase plots of the steady period response at a boundary critical AC voltage (5.3 V): AC symmetry breaking and pull–in.

## Data Availability

Not applicable.

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
