# Peer review of "Enhancing the Recognition Task Performance of MEMS Resonator-Based Reservoir Computing System via Nonlinearity Tuning"

_micromachines, 2022, doi:10.3390/mi13020317_

Round 1
Reviewer 1 Report
Dear Athors,
The manuscript “Enhancing the Recognition-task Performance of MEMS Resonator-based Reservoir Computing System via Nonlinearity Tuning” is a continuation of the studies [25] (Microsystems & Nanoengineering 2021, 7, 83, 351 doi:10.1038/s41378-021-00313-7). The results of the search for optimal operating condition to achieve higher accuracy and faster computing speed for the RC hardware are presented. The main results (the influence of the quality factor of the resonator on the accuracy of recognition and the determination of optimal operating point near the edge of chaos) are quite sufficient for publication in a highly rated journal.
But there are several questions about the presentation of these results in the manuscript.
- The quality factor of the resonator is varied by the pressure in the chamber.The manuscript does not present the dependence of the quality factor on this pressure, although you position the research as experimental.An appropriate figure is needed.
- Howisformula (1) obtained ?This formula is not in the suggested reference [26].What values of the parameters Co and d were used in the simulations?
- You can present the SFD model in more detail (paragraph 3.1 can be expanded), since this is an important part of the methodology of your study.I could not find reference [30] (is there another source?).
- A more detailed explanation of Figure 5a (the abscissa coefficient ? ) and a few words about the scenario of the transition to chaos in the Duffing system (1), (4) is needed (expand paragraph 4.2).
- In Introduction at the end of the first paragraph (line 52), you can add references to reservoir computing using single oscillators [x] and discrete mappings [xx]. For example:
[x] M. R. Shougat, X. Li, T. Mollik, E. Perkins A Hopf physical reservoir computer Scientific Reports, 2021, V. 11, article № 19465, doi: 10.1038/s41598-021-98982-x.
[xx] A. Velichko, Neural Network for Low-Memory IoT Devices and MNIST Image Recognition Using Kernels Based on Logistic Map, Electronics 2020, 9, 1432, doi:10.3390/electronics9091432.
Summary. I invite the authors of this article to answer my questions and make a minor revision оf manuscript. Sorry, for my English.
Reviewer 2 Report
This manuscript reports a piece of work on reservoir computing in neuromorphic paradigm that employs nonlinearity turning of a MEMS resonator. After careful reading and checking the soundness of the content, the reviewer finds that the manuscript is not suitable for publication in its current format. First of all, to this reviewer, the major work in the manuscript is focused on the algorithms for pattern recognition accuracy and speed, which is off from the subject of this journal. Secondly, the MEMS resonator part in the work, which is the only content that’s closely related to “micromachine”, is barely presented. Through the entire manuscript, there is only a small graph (an inset in Fig. 1.a) showing an SEM image of the MEMS resonator, and a driving frequency of 189.5 kHz (Line 128) is mentioned in the text. No other dimensional, static and dynamic parameters of the resonator are given (Line 95 to Line 99, Line 194 to 195) to justify the feasibility and reliability in using such a MEMS model. Thirdly, the presentation is so tedious and full of long, disconnected sentences and clauses, that many important points the authors want to make are hard to understand. Based on the above observations, this reviewer suggests that the manuscript be rejected from this journal. It’s suggested that this work be submitted to another journal in neural network area, after the authors aggressively reorganize the manuscript and streamline the presentation.Author Response
Please see the attachment.

Reviewer 3 Report
This work presents a hybrid nonlinear combinatorial modulation strategy to search the optimal operating condition to achieve higher accuracy and faster computing speed for the hardware RC. The topic is relatively new and should be of interest to MEMS researchers. Below are some of my comments;
- In Figure 1, it will be more appropriate if authors can add a clear SEM of the fabricated resonator
- On line 111 reference is missing
- The English language style and sentence structuring should be improved.
Round 2
Reviewer 2 Report
With necessary information being added, the nonlinear dynamics and the tuning mechanism of the resonator can now be better understood for the purpose of recognition performance improvement. The soundness of the scientific base is thus justified to this reviewer.
There are still some editing work to be done. For instance, Fig. 3 has some repeated subplots...
The reviewer suggests that the manuscript be accepted for publication per authors' and editor's further rigorous editing.